# IgLON5-IgG: Innocent Bystander or Perpetrator?

**DOI:** 10.3390/ijms25147956

**Published:** 2024-07-21

**Authors:** Jane Andersen, Bronte Jeffrey, Winny Varikatt, Michael Rodriguez, Ming-Wei Lin, David A. Brown

**Affiliations:** 1Department of Immunology, NSW Health Pathology-ICPMR, Westmead Hospital, Sydney, NSW 2145, Australia; bronte.jeffrey@gmail.com (B.J.); mingwei.lin@health.nsw.gov.au (M.-W.L.); david.brown1@health.nsw.gov.au (D.A.B.); 2Faculty of Medicine and Health, The University of Sydney, Sydney, NSW 2050, Australia; winny.varikatt@health.nsw.gov.au (W.V.); michael.rodriguez@health.nsw.gov.au (M.R.); 3Faculty of Medicine, Western Sydney University, Sydney, NSW 2751, Australia; 4Department of Tissue Pathology and Diagnostic Oncology, NSW Health Pathology-ICPMR, Westmead Hospital, Sydney, NSW 2145, Australia; 5Douglass Hanly Moir Pathology, Sydney, NSW 2000, Australia

**Keywords:** IgLON5, anti-IgLON5-associated disease, IgLON5-IgG, neurodegeneration, autoimmune encephalitis

## Abstract

Anti-IgLON5 (IgLON5-IgG)-associated disease is a newly defined clinical entity. This literature review aims to evaluate its pathogenesis, which remains a pivotal question. Features that favour a primary neurodegenerative mechanism include the non-inflammatory tauopathy neuropathological signature and overrepresentation of microtubule-associated protein tau (*MAPT*) H1/H1 genotype as seen in other sporadic tauopathies. In contrast, the cell-surface localisation of IgLON5, capability of anti-IgLON5 antibodies to exert direct in vitro pathogenicity and disrupt IgLON5 interactions with its binding partners, human leukocyte antigen (HLA)-DRB1*10:01 and HLA-DQB1*05:01 allele preponderance with high affinity binding of IgLON5 peptides, and responsiveness to immunotherapy favour a primary autoimmune process. The presentation and course of anti-IgLON5-associated disease is heterogenous; hence, we hypothesise that a multitude of immune mechanisms are likely simultaneously operational in this disease cohort.

## 1. Introduction

Anti-IgLON5 antibodies (IgLON5-IgG) from serum and cerebrospinal fluid (CSF) were first described clinically in a cohort of patients with sleep apnoea, non-rapid eye movement (non-REM) and REM parasomnias, and stridor in 2014 [1]. In the years following this seminal study, there has been increasing research into this disease, albeit largely limited to case reports and case series given its estimated incidence of 1 in 150,000 [2]. The IgLON family is a group of five cell adhesion molecules (IgLON1-5) each with three immunoglobulin-like domains, which mediate a multitude of cellular interactions, particularly at the blood–brain barrier (BBB) [3,4,5,6]. An investigation of the molecular evolution of the IgLON family suggests a role in regulating neural growth and complexity by possessing several motifs that may influence cellular migration and proliferation as well as BBB permeability [7]. While BBB dysfunction, characterized by an increased albumin quotient (CSF albumin/serum albumin ratio) and total CSF protein, appears to be a feature in approximately half of patients with IgLON5-IgG disease, the physiologic role of IgLON5 in BBB integrity is incompletely characterized [8,9]. It is unclear whether IgLON5-IgG is synthesized intrathecally, peripherally, or a combination of both [10,11]. The pathogenesis of IgLON5-IgG disease remains a pivotal question. To date, the literature has focused on whether IgLON5-IgG disease is a primary neurodegenerative process with a secondary inflammatory response against IgLON5 or a primary autoimmune process driven by pathogenic IgLON5-IgG with subsequent neurodegeneration. The aim of this review article is to synthesise the existing literature and evaluate this question. We hypothesise that a multitude of immune mechanisms are likely simultaneously operational in IgLON5-IgG disease as evidenced by the heterogeneity of the patient cohort and, thus, we caution against relying on a dichotomous classification of the pathogenesis.

## 2. Evidence for a Primary Neurodegenerative Mechanism

The following evidence favours a primary neurodegenerative mechanism underlying IgLON5-IgG disease. Firstly, the neuropathological signature of IgLON5-IgG disease as first described by Sabater et al. and formalised in the recently proposed IgLON5-IgG disease neuropathological criteria involves three key features: (1) subcortical distribution of the tau pathology predominantly affecting the hypothalamus, brainstem tegmentum, and upper spinal cord; (2) tau pathology, nearly exclusively neuronal, with little or no glial and white matter involvement; and (3) disease-associated tau composed of both 3R and 4R isoforms, which are defined by the presence of three and four microtubule-binding repeats, respectively [1,12]. The topographical distribution of the tau pathology is similar to that seen in another tauopathy, progressive supra-nuclear palsy (PSP); however, the absence of glial pathology and sparse supratentorial and basal ganglia involvement in IgLON5-IgG disease is distinct [12,13]. Fearnley et al. examined the spatiotemporal expression of all IgLON family members in the developing murine nervous system at prenatal and postnatal stages [14]. IgLON5 expression was detected prenatally in the eye and olfactory system with subsequent postnatal downregulation at both sites, throughout development in the cerebral cortex, hippocampus, entorhinal region, habenula, and various nuclei of the thalamus and postnatally in the cerebellum and spinal cord [14]. This contrasts against postmortem pathology-mapping, which is largely based on end-stage pathology such as neuronal loss; partial overlap with the regions of interest identified by murine modelling of IgLON5 expression is noted and may explain at least part of the regional vulnerability observed in IgLON5-IgG disease [1,14,15]. The tau filaments seen in IgLON5-IgG disease are of similar structure to those in Alzheimer’s disease (AD), appearing ultrastructurally as paired helical filaments, being composed of both 3R and 4R tau isoforms [12,16].

Secondly, the microtubule-associated protein tau (*MAPT*) H1/H1 genotype is significantly overrepresented in patients with IgLON5-IgG disease (81.5%, 22/27) compared to healthy controls (46.5%, 54/116) (OR 5.05, 95% CI 1.79–14.25, *p* = 0.0007) [17]. The *MAPT* locus is divided into two haplotypes (H1 and H2) by a common inversion polymorphism [18,19,20]. It is well evidenced that H1 homozygosity is associated with an increased risk for tauopathies like PSP, corticobasal degeneration (CBD), and AD [19,21,22,23,24,25,26]. The underlying mechanism linking the H1 haplotype with neurodegeneration is likely related to altered expression levels, altered splicing, or a combination of both [25]. Indeed, the H1 haplotype has been associated with an increased expression of *MAPT* overall, as well as increased expression of the key exon 10, which, through alternative splicing, generates six tau isoforms (three 3R isoforms and three 4R isoforms) whose expression ratio tightly directs microtubule dynamics [25,26,27,28,29,30]. It is important to note that these findings have not yet been evaluated in an IgLON5-IgG disease cohort.

Finally, poor response to immunotherapy was initially reported and suggested a primary neurodegenerative aetiology of IgLON5-IgG disease [1,31]; however, more recent evidence is discordant with this [2,8,32]. This can likely be attributed to earlier studies having limited case numbers as well as cohort selection bias given that patients were recruited based on a narrow clinical phenotype (the originally described sleep disorder). Subsequent studies have had larger cohorts and selection based on serological screening of stored samples, thus revealing a more diverse spectrum of clinical phenotypes. Further studies of treatment efficacy in populous and clinically diverse cohorts are essential to comprehensively understand IgLON5-IgG disease.

## 3. Evidence for a Primary Autoimmune Mechanism

The following key principles favour a primary autoimmune aetiology of IgLON5-IgG disease. The cell-surface localisation and role of IgLON5 in the dynamic interactions that occur at the BBB implicate it as a biologically plausible and accessible central nervous system (CNS) autoantigen [1,33,34]. Furthermore, IgLON5-IgG in patient sera exerts a directly pathogenic effect on cultured neurons by causing an irreversible decrease in the cell-surface density of IgLON5 through internalisation, reduction in electrical neuronal activity, and increased frequency of degenerative changes in neurons, such as axonal blebbing and fragmentation [34,35]. Purification of IgG subclasses revealed that this effect was specifically attributable to IgG1 antibodies and not IgG4 antibodies [34]. Interestingly, however, the IgG4 subclass was predominant in the majority of patients from this study, accounting for, on average, 64% of total IgLON5-IgG isolated from patient sera compared to IgG1 antibodies accounting for, on average, 33% of total IgLON5-IgG [34]. It is well established that IgG1 antibodies engage C1q and FcγR more efficiently compared to the reduced binding affinity shown by IgG4 antibodies, and thus, IgG1 antibodies are more potent triggers of pro-inflammatory effector mechanisms; however, IgG4 antibodies are also capable of exerting a pathogenic effect as exemplified in Myasthenia gravis where IgG4 antibodies mediate receptor inactivation [36,37]. In keeping with this, Landa, et al. identified another potential mechanism by which IgLON5-IgG isolated from patient sera may exert a pathogenic effect [38]. Firstly, they demonstrated that physiologic IgLON5 undergoes spontaneous ectodomain shedding and interacts with other members of the IgLON family and, secondly, that both IgLON5-IgG1 and IgG4 subclasses disrupt this interaction [38]. A recent autopsy series published by Berger-Sieczkowski et al. identified two patients with short disease duration without the typical anti-IgLON5-related tauopathy who had extensive neuropil deposition of IgLON5-IgG4 in the brainstem tegmentum, olivary nucleus, and cerebellar cortex [15]. These IgG4 deposits were accompanied by lesser amounts of IgG1 [15]. We postulate that both IgLON5-IgG1 and IgLON5-IgG4 are operational through different mechanisms, thus, potentially explaining a degree of the clinical heterogeneity observed in this disease. There are currently no studies published that investigate complement activation in IgLON5-IgG disease; however, this would be useful in further distinguishing the IgG1- and IgG4-mediated processes in this condition.

Human leukocyte antigen (HLA) genotyping of patients with IgLON5-IgG disease highlighted a genetic susceptibility for autoimmune disease. Initial works identified a robust association with the HLA-DRB1*10:01 and HLA-DQB1*05:01 alleles, reported at a frequency of 57.1–100% of patients [1,17,32]. In addition, IgLON5 peptides exhibit high affinity for the HLA-DRB1 molecules (DRB1*01:01, DRB1*10:01, and DRB1*09:01) isolated from patients [17]. A recent study by Yogeshwar et al. was the largest HLA-association analysis in IgLON5-IgG disease to date, and their findings strongly supported that HLA-DQ, and not HLA-DR, is the actual determinant for disease risk [39]. HLA class II molecules are known to play an integral role in antigen presentation to activate the adaptive immune response, including B-cells and T-cells [17]. The weighting of these responses, based on genetic variation such as HLA, seems likely to impact the individual presentation of IgLON5-IgG disease.

Current evidence supports IgLON5-IgG disease responsiveness to immunotherapy [8,32]. A systematic review by Cabezudo-García et al. reported promising responses and sustained response rates with azathioprine (AZA) (100% [5/5]) and mycophenolate mofetil (MMF) (75% [3/4]), albeit always administered in combination with other agents like corticosteroids (CS) (CS alone had a response rate of 34.2% [12/35]) [32]. A large 2023 retrospective cohort study of 53 patients by Grüter et al. reported favourable response of IgLON5-IgG disease to periodic intravenous immunoglobulin (IVIg) (86.7% [13/15]), AZA (85.7% [6/7]), repetitive plasma exchange (PLEX) (75% [3/4]), and rituximab (RIX) (72.2% [13/18]).

Finally, a notable mention is made of several patients’ case reports with clinical and serological evidence of IgLON5-IgG disease without degenerative features like hyperphosphorylated tau in brain tissue, which may shed light on the sequence of pathogenesis of this disease entity, specifically, that tau accumulation occurs later in the disease course and is a consequence of antibody-mediated neuronal dysfunction [15,40,41,42,43].

## 4. Heterogeneity of the IgLON5-IgG Disease Cohort

The presentation and progress of IgLON5-IgG disease is diverse. Subgroups appear to diverge in terms of disease time-course and symptomatology [2,8,44]. Grüter et al. reported 28% (15/53) of patients had subacute disease onset (≤4 weeks) while 72% (38/53) had slow progressive disease onset (>4 weeks) with (32%, 12/38) or without (68%, 26/38) overlapping relapse-like exacerbations [8]. Interestingly, patients with subacute disease onset were significantly more likely to manifest psychosis and/or hallucinations as part of their clinical phenotype, have an inflammatory CSF characterised by pleocytosis (albeit mild), and exhibit a more pronounced response to immunotherapy [8]. An important confounding factor is that patients with subacute disease onset, more in keeping with a traditional autoimmune picture, received significantly earlier diagnosis leading to earlier initiation of treatment: an independent predictor of a favourable prognostic outcome [8]. Similar observations were also reported by Gaig et al. who noted that 24% (17/72) of their IgLON5-IgG disease cohort had subacute disease onset (<4 months) and by Honorat et al. who noted that 25% (5/20) of their cohort evolved symptoms in a subacute manner (<4 weeks) [2,44]. Neither clinical characteristics nor CSF examination stratified specifically by disease time-course were available in either of these studies. Furthermore, the IgLON5-IgG disease cohort may be stratified by HLA-DRB1*10:01 and HLA-DQB1*05:01 positivity. Grüter et al. identified that HLA-DRB1*10:01- and HLA-DQB1*05:01-positive patients were significantly younger at disease onset, more frequently exhibited characteristic sleep disorders, and had a higher IgLON5-IgG titre compared to HLA-DRB1*non-10:01 and HLA-DQB1*non-05:01 patients [8]. Similarly, Gaig et al. reported that HLA-DRB1*10:01-positive patients were younger (median 64.5 years [range 46–77]) at diagnosis and more frequently presented with sleep or bulbar clinical phenotypes compared to HLA-DRB1*non-10:01 patients who were older (median 71 years [range 61–83]) at diagnosis and tended to present with PSP-like or cognitive impairment phenotypes; all of these findings reached statistical significance [17]. As described earlier, we further postulate that IgG1 or IgG4 subclass antibodies may influence whether that patient’s disease is more pro-inflammatory and subacute or non-inflammatory and chronic, respectively. Ultimately, the model we propose is that patients with IgLON5-IgG disease may exhibit features of either a primarily neurodegenerative process, a primarily autoimmune process, or a combination of both. For example, a chronic onset process of neurodegeneration presenting with a specific clinical phenotype may then be augmented by exposure to an inflammatory insult, such as a virus infection, triggering an autoimmune process and altering disease course. The heterogeneity of IgLON5-IgG disease is complex and further observational cohort studies are essential.

## 5. An Atypical Case and Future Directions

Finally, we revisit a previously published atypical case assessed at our centre, initially reported in 2018 [42]. A 49-year-old man presented with a two-year history of cold intolerance followed by the development of involuntary jerking movements, impaired sleep, gait ataxia, dysarthria, and cognitive decline. He was systemically well throughout this chronic course and had no significant past medical history. He was investigated extensively. Prior to the result of his serum and CSF positivity for IgLON5-IgG becoming available, he underwent a stereotactic brain biopsy. This demonstrated cuffing of the white matter blood vessels with CD3+ T lymphocytes as well as mild leptomeningeal chronic inflammation with cortical and white matter gliosis and microglial activation (Figure 1) [42]. Neurodegenerative stains (tau, α-synuclein, β-amyloid, phosphorylated-TDP43, and P62) were all negative; however, it is worth noting that the superficial areas biopsied (cortex and cerebellum) usually do not demonstrate tauopathy in this disease.

The initial treatment regimen consisted of RIX, alemtuzumab (ALEM), PLEX, IVIg, and CS in close succession. In addition to the course of ALEM and RIX, he received two separate courses of intravenous cyclophosphamide (CP) (each course consisting of 500 mg fortnightly for six doses) as per the EuroLupus protocol [45]. He remains on monthly PLEX and IVIg as well as MMF. ALEM was selected based on the presence of CD3+ T lymphocytes on brain biopsy (Figure 1), and to the best of our knowledge, this is the only report of its use to treat IgLON5-IgG disease [42]. ALEM, a monoclonal antibody that targets the CD52 antigen abundantly expressed on T-lymphocytes, has proven useful in the treatment of T-cell-driven pathologies like haematologic malignancies [46]. It was previously thought that T lymphocytes do not have a role in the pathogenesis of IgLON5-IgG disease [17]. However, the usefulness of alemtuzumab in this case and similar reports of perivascular and parenchymal CD3+ and CD8+ T lymphocytes in IgLON5-IgG disease autopsy specimens may contradict this view and support the consideration of a more personalised approach to diagnosis and treatment of rare diseases such as IgLON5-IgG disease [15,42].

Careful phenotyping of the individual patient’s likely underlying pathogenesis, as demonstrated in this atypical case, is essential to instituting the most appropriate treatment regimen. A crucial future direction of this field will be further phenotyping of the IgLON5-IgG disease cohort, in particular, further characterisation of the subsets of patients who present with a more traditional autoimmune picture and those with a more gradual onset of disease, perhaps more in keeping with a neurodegenerative phenotype. The former is indicated by evidence of inflammation and a more acute to subacute presentation with a propensity for the characteristic sleep disorder phenotype. The subset of patients in keeping with a neurodegenerative picture generally present in a more non-inflammatory and chronic manner with a propensity for the PSP-like and cognitive impairment phenotypes [8,17]. While this carries a clear clinical implication for accurate and timely diagnosis, at this stage, the treatment pathway remains equivalent between the subgroups. Immunotherapy is the cornerstone for addressing the inflammatory component of IgLON5-IgG disease as there are currently no effective treatments for the neurodegenerative component though putative therapeutic targets have been identified in the context of other tauopathies [47,48]. Interestingly, these have a role in inflammation as well, indicating that future diagnostic approaches such as sequencing/proteomics may help stratify patient therapy [47,48]. Treatments targeting pathological tau are predominantly in the discovery and preclinical stages [48].

## 6. Conclusions

In conclusion, IgLON5-IgG disease is a relatively newly defined clinical entity. The non-inflammatory tauopathy neuropathological signature and overrepresentation of MAPT H1/H1 genotype as seen in other sporadic tauopathies is consistent with a primary neurodegenerative process. In these cases, it is conceivable that neurodegeneration preceded the development of an antibody response. In contrast, the cell-surface localisation of IgLON5, capability of IgLON5-IgG to exert direct in vitro pathogenicity and disrupt IgLON5 interactions with its binding partners, HLA-DRB1*10:01 and HLA-DQB1*05:01 allele preponderance loaded with high affinity binding of IgLON5 peptides, and responsiveness to immunotherapy favour a primary autoimmune mechanism.

Nonetheless, we caution against relying on this dichotomous classification. We hypothesise that a multitude of immune mechanisms are likely simultaneously operational in response to varied triggering factors as evidenced by the time-course and phenotype heterogeneity of the IgLON5-IgG disease cohort. This heterogeneity appears to be explained, in part, when patients are stratified by HLA-DRB1*10:01 and HLA-DQB1*05:01 positivity. HLA class II molecules are implicated in antigen presentation and the subsequent activation of the adaptive immune response, including B-cells and T-cells. The weighting of these responses, based on genetic variation such as HLA, seems likely to impact the individual presentation of IgLON5-IgG disease. We promote moving toward a personalised approach to diagnosis and treatment, one that encapsulates the factors outlined above to best define the underlying dominant pathogenic factors in the individual. Moreover, careful clinical phenotyping and bio-banking of these rare diseases is encouraged so that further hypotheses may be generated and tested.

## Figures and Tables

**Figure 1 ijms-25-07956-f001:**
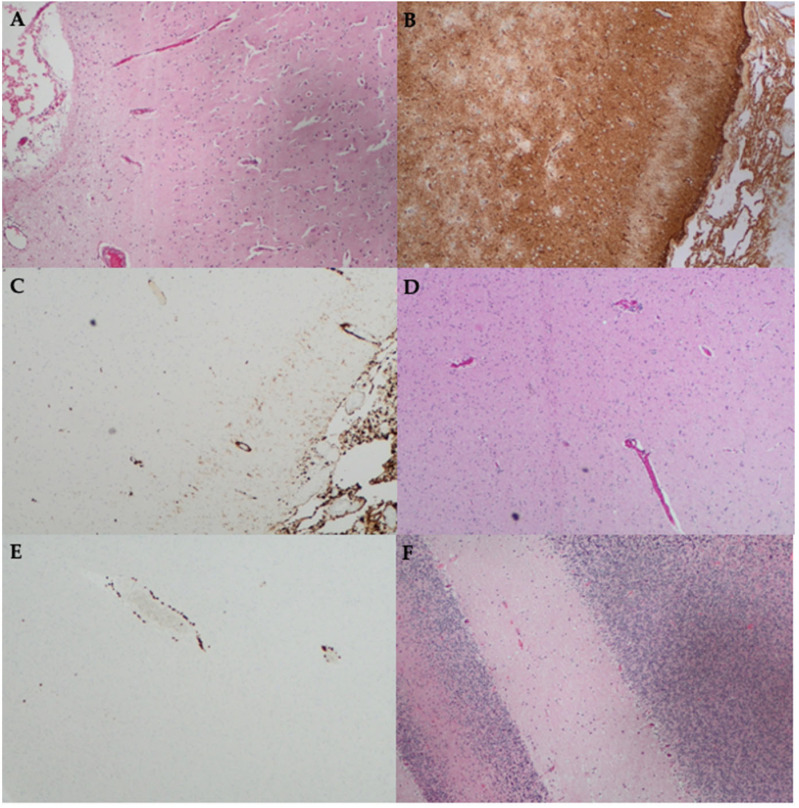
Histopathology findings from the stereotactic brain biopsy of a patient with IgLON5-IgG disease, which was previously published as a case report [42]. Biopsy of the right frontal cortex stained with (**A**) Hematoxylin and Eosin (H&E) stain shows leptomeningeal oedema; (**B**) Glial fibrillary acidic protein (GFAP) shows gliosis; (**C**) CD163 shows increased histiocytes in the leptomeninges and very mild increase in the underlying cortex; (**D**) H&E stain shows subtle perivascular lymphocytic infiltrate; and (**E**) CD3 immunohistochemistry shows CD3+ T lymphocyte cuffing of blood vessels. (**F**) Biopsy of the left cerebellum stained with H&E shows a reduction in Purkinje cells. Magnification ×100 for all.

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
