# Peer review of "IgLON5-IgG: Innocent Bystander or Perpetrator?"

_ijms, 2024, doi:10.3390/ijms25147956_

Round 1

Reviewer 1 Report

Comments and Suggestions for Authors

Andersen et al. delves into the pathogenesis of IgLON5-IgG associated disease, a newly defined clinical entity. The authors explore the intricate balance between neurodegenerative and autoimmune mechanisms underlying this condition. They discuss the neuropathological signature of IgLON5-IgG disease, highlighting features suggestive of a primary neurodegenerative process, such as tau deposition and MAPT H1/H1 genotype overrepresentation. Additionally, the review delves into the autoimmune aspects of the disease, including HLA allele associations and responsiveness to immunotherapy.

The authors caution against a simplistic dichotomous classification of the pathogenesis of IgLON5-IgG associated disease, emphasizing the heterogeneity in patient presentation and the likelihood of multiple immune mechanisms at play simultaneously. They advocate for a personalized approach to diagnosis and treatment, considering individual factors to define the dominant pathogenic processes. Furthermore, the review underscores the importance of careful clinical phenotyping and biobanking to generate and test hypotheses for this rare disease.

Overall, this review provides a comprehensive analysis of IgLON5-IgG associated disease, offering insights into its complex pathogenesis and advocating for a nuanced understanding that integrates both neurodegenerative and autoimmune perspectives.

The authors present a well-structured and coherent analysis of the current understanding of IgLON5-IgG associated disease, incorporating relevant literature to support its arguments. The authors effectively highlight the key features favoring a primary neurodegenerative mechanism, such as the non-inflammatory tauopathy neuropathological signature and the overrepresentation of the MAPT H1/H1 genotype. Additionally, the discussion on the potential autoimmune processes, including HLA allele preponderance and responsiveness to immunotherapy, adds depth to the analysis and contributes to the ongoing discourse in the field.

a. Provide a more detailed discussion on the implications of the MAPT H1/H1 genotype overrepresentation in IgLON5-IgG associated disease compared to other tauopathies. Consider elucidating the specific genetic mechanisms that may underlie this association.

b. Include a section addressing the clinical implications of the proposed neurodegenerative and autoimmune mechanisms in the diagnosis and management of IgLON5-IgG associated disease. This could enhance the practical relevance of the review for clinicians and researchers.

c. Expand on the discordant evidence regarding the initial poor response to immunotherapy in IgLON5-IgG associated disease. Discuss potential factors contributing to this discrepancy and propose future research directions to reconcile these findings.

d. Incorporate a comparative analysis (if any) of the neuropathological features of IgLON5-IgG associated disease with other neurodegenerative and autoimmune conditions to provide a broader context for understanding its pathogenesis.

e. Consider including a section on emerging therapeutic strategies targeting the neurodegenerative and autoimmune components of IgLON5-IgG associated disease. This could offer valuable insights into potential treatment avenues and research priorities in the field.

Author Response

Thank you very much for taking the time to review this manuscript. We appreciate your feedback. Please find the detailed responses below and the corresponding revisions in track changes in the re-submitted files

Comments 1: Provide a more detailed discussion on the implications of the MAPT H1/H1 genotype overrepresentation in IgLON5-IgG associated disease compared to other tauopathies. Consider elucidating the specific genetic mechanisms that may underlie this association.

Response 1: Section 2 ‘Evidence for a primary neurodegenerative mechanism’ (page 2, lines 65-80) has been expanded to clearly identify the parallel between MAPT (microtubule associated tau protein) H1/H1 overrepresentation in IgLON5-IgG disease as seen in other tauopathies like corticobasal degeneration, progressive supranuclear palsy, and Alzheimer's disease. We note that the underlying mechanism linking the H1 haplotype with neurodegeneration is likely related to altered expression levels, altered splicing, or a combination of both.

Comments 2: Include a section addressing the clinical implications of the proposed neurodegenerative and autoimmune mechanisms in the diagnosis and management of IgLON5-IgG associated disease. This could enhance the practical relevance of the review for clinicians and researchers.

Response 2: Section 5 ‘An atypical case and future directions’ (pages 4-5, lines 433-488) has been expanded in response to this comment. We outline that the autoimmune and neurodegenerative patient subgroups appear to have a distinct propensity for presenting phenotype (autoimmune – sleep disorder; neurodegenerative – PSP-like and cognitive impairment) and that this is relevant for accurate and timely diagnosis. We further outline that, at this stage, the treatment pathway remains equivalent between the subgroups. Specifically, immunotherapy is the cornerstone to address the inflammatory component of IgLON5-IgG disease as there are currently no effective treatments for the neurodegenerative tauopathy component; however, putative therapeutic targets have been identified and treatments targeting pathological tau remain predominantly in the discovery and preclinical stages.

Comments 3: Expand on the discordant evidence regarding the initial poor response to immunotherapy in IgLON5-IgG associated disease. Discuss potential factors contributing to this discrepancy and propose future research directions to reconcile these findings.

Response 3: We have edited section 2 ‘Evidence for a primary neurodegenerative mechanism’ (page 2, lines 81-89) to provide a clearer discussion around the initially reported refractory nature of IgLON5-IgG disease to immunotherapy. We suggest that the primary reasons for this are likely initial limited case numbers as well as cohort selection bias as early cohorts were recruited based on a narrow clinical phenotype (the originally described sleep disorder) while subsequent studies had selection based on serological screening of stored samples, thus, revealing a more diverse spectrum of clinical phenotypes.

Comments 4: Incorporate a comparative analysis (if any) of the neuropathological features of IgLON5-IgG associated disease with other neurodegenerative and autoimmune conditions to provide a broader context for understanding its pathogenesis.

Response 4: Section 2 ‘Evidence for a primary neurodegenerative mechanism’ (page 2, lines 54-65) now includes a comparison of the neuropathological signature of IgLON5-IgG disease and other tauopathies. Specifically, we discuss how the topographical distribution of the tauopathy is similar to progressive supra-nuclear palsy and that the composition of the disease-associated tau is similar to Alzheimer’s disease.

Comments 5: Consider including a section on emerging therapeutic strategies targeting the neurodegenerative and autoimmune components of IgLON5-IgG associated disease. This could offer valuable insights into potential treatment avenues and research priorities in the field.

Response 5: In section 3 ‘Evidence for a primary autoimmune mechanism’,  (page 3, lines 341-348) we discuss the most current therapeutic strategies for IgLON5-IgG disease treatment, namely the use of immunotherapeutics in combination for a more efficacious response. As mentioned in response to comments 2, we also include a statement on the lack of current available therapeutics for the tauopathy component of this disease but do highlight that putative therapeutic targets have been identified and that treatments targeting pathological tau remain predominantly in the discovery and preclinical stages (Section 5 ‘An atypical case and future directions’, page 5, lines 480-488).

Reviewer 2 Report

Comments and Suggestions for Authors This review paper mainly focused on the neurodegenerative versus autoimmune theories regarding the pathogenesis of IgLON5-related diseases. Evidence for and against each theory was collated. My opinion is that these two theories are not mutually exclusive, and several issues could be further addressed in this review. My comments are as follows.    1. The authors have discussed about the implications of different IgG subclasses in the pathogenesis of anti-IgLON5 disease. I would suggest that the authors also discuss about the role of complement activation (which may serve to distinguish IgG1- from IgG4-mediated processes) in the pathogenesis of this condition.    2. The authors mentioned about the possible physiological role of IgLON5 at the blood brain barrier. Does anti-IgLON5 IgG have any effect on the blood brain barrier?    3. Are anti-IgLON5 antibodies produced intrathecally, or are they produced at periphery and somehow transferred into the CNS?   4. Regarding autoimmune theory, how about the roles of B cell versus that of T cell?    5. I would suggest the authors incorporate the following reference for an update: HLA-DQB1*05 subtypes and not DRB1*10:01 mediates risk in anti-IgLON5 disease. Yogeshwar SM, Muñiz-Castrillo S, Sabater L, Peris-Sempere V, Mallajosyula V, Luo G, Yan H, Yu E, Zhang J, Lin L, Fagundes Bueno F, Ji X, Picard G, Rogemond V, Pinto AL, Heidbreder A, Höftberger R, Graus F, Dalmau J, Santamaria J, Iranzo A, Schreiner B, Giannoccaro MP, Liguori R, Shimohata T, Kimura A, Ono Y, Binks S, Mariotto S, Dinoto A, Bonello M, Hartmann CJ, Tambasco N, Nigro P, Prüss H, McKeon A, Davis MM, Irani SR, Honnorat J, Gaig C, Finke C, Mignot E. Brain. 2024 Mar 1:awae048. doi: 10.1093/brain/awae048.   6. Some statements are redundant. For example, "we caution against relying on a dichotomous classification ...." are repeatedly stated in the Introduction and Conclusion. Comments on the Quality of English Language

no specific comment

Author Response

Thank you very much for taking the time to review this manuscript. We appreciate your feedback. Please find the detailed responses below and the corresponding revisions in track changes in the re-submitted files.

Comments 1: The authors have discussed the implications of different IgG subclasses in the pathogenesis of anti-IgLON5 disease. I would suggest that the authors also discuss about the role of complement activation (which may serve to distinguish IgG1- from IgG4-mediated processes) in the pathogenesis of this condition. 

Response 1: There are currently no studies published investigating complement activation in IgLON5-IgG disease. This has now been clearly stated in section 3 ‘Evidence for a primary autoimmune mechanism’ in order to highlight it as a priority for future research (page 3, lines 328-330). We have additionally highlighted the differences in FcR engagement between IgG1 and IgG4, which may also modify pathological course of IgG mediated diseases (page 3, lines 315-318).

Comments 2: The authors mentioned about the possible physiological role of IgLON5 at the blood brain barrier. Does anti-IgLON5 IgG have any effect on the blood brain barrier?

Response 2: There is evidence of BBB dysfunction in approximately half of patients with IgLON5-IgG disease characterised by increased serum/CSF albumin quotient and increased total CSF protein; however, the physiologic role of IgLON5 in BBB integrity is incompletely characterized. Section 1 ‘Introduction’ has been updated to reflect this (page 1, lines 35-38).

Comments 3: Are anti-IgLON5 antibodies produced intrathecally, or are they produced at periphery and somehow transferred into the CNS?

Response 3: It remains unclear whether IgLON5-IgG is synthesized intrathecally, peripherally, or a combination of both. This statement is now represented in section 1 ‘Introduction’ (page 1, lines 38-39).

Comments 4: Regarding autoimmune theory, how about the roles of B cell versus that of T cell?

Response 4: Section 3 ’Evidence for a primary autoimmune mechanism’ (page 3, lines 345-348) highlights the principle that the association of IgLON5-IgG disease with HLA class II molecules implicates antigen presentation to activate the adaptive immune response, including B-cells and T-cells. Furthermore, the weighting of these responses, based on genetic variation such as HLA, seems likely to impact on the individual presentation of IgLON5 disease. We additionally emphasise this point in the conclusion (page 6, lines 551-554) to tie together the concept for researchers and clinicians. We further discuss the role of B-cell and T-cell directed therapies in the form of Alemtuzumab in section 5 ’An atypical case and future directions’ (page 4, lines 428-438) based on the presence of CD3+ T-cells on stereotactic brain biopsy.

Comments 5: I would suggest the authors incorporate the following reference for an update: HLA-DQB1*05 subtypes and not DRB1*10:01 mediates risk in anti-IgLON5 disease. Yogeshwar SM, Muñiz-Castrillo S, Sabater L, Peris-Sempere V, Mallajosyula V, Luo G, Yan H, Yu E, Zhang J, Lin L, Fagundes Bueno F, Ji X, Picard G, Rogemond V, Pinto AL, Heidbreder A, Höftberger R, Graus F, Dalmau J, Santamaria J, Iranzo A, Schreiner B, Giannoccaro MP, Liguori R, Shimohata T, Kimura A, Ono Y, Binks S, Mariotto S, Dinoto A, Bonello M, Hartmann CJ, Tambasco N, Nigro P, Prüss H, McKeon A, Davis MM, Irani SR, Honnorat J, Gaig C, Finke C, Mignot E. Brain. 2024 Mar 1:awae048. doi: 10.1093/brain/awae048.

Response 5: Thank you for this suggestion. We have updated section 3 ’Evidence for a primary autoimmune mechanism’ to include this reference (page 3, lines 341-344).

Comments 6: Some statements are redundant. For example, "we caution against relying on a dichotomous classification ...." are repeatedly stated in the Introduction and Conclusion.

Response 6: We have revised the manuscript to ensure it is succinct. Specifically regarding the phrase ‘dichotomous classification’, we have removed this from the abstract and it now appears only in the introduction and conclusion for a cohesive summary of one of the key messages of this review.

Round 2

Reviewer 2 Report

Comments and Suggestions for Authors 1. Please recheck the following sentence (page 1, line 37): "...... characterised by increased serum/CSF albumin quotient and total CSF protein ......".   2. The Ref.2 is incomplete.   3. How to interpret the various movement disorders in anti-IgLON5 disease in the context of autoimmune or neurodegeneration theories?   4. Ref 8 and Ref 31 are redundant.  Comments on the Quality of English Language

Minor editing of English language required

Author Response

Thank you very much for taking the time to review this manuscript. We appreciate your feedback. Please find the detailed responses below and the corresponding revisions in track changes in the re-submitted files.

Comments 1: Please recheck the following sentence (page 1, line 37): "...... characterised by increased serum/CSFalbumin quotient and total CSF protein ......".

Response 1: Thank you for your edit. This sentence has been corrected to “albumin quotient (CSF albumin/serum albumin ratio)” (page 1, line 36).

Comments 2: The Ref.2 is incomplete.   

Response 2: Reference 2 was related to the estimated incidence rate of anti-IgLON5 disease; however, upon review, it was not the primary paper stating this estimate. The manuscript has been revised to reference the incidence rate to the original paper (Honorat JA, Komorowski L, Josephs KA, Fechner K, St Louis EK, Hinson SR, et al. IgLON5 antibody: Neurological accompaniments and outcomes in 20 patients. Neurol Neuroimmunol Neuroinflamm. 2017;4(5):e385.) (page 1, line 30).

Comments 3: How to interpret the various movement disorders in anti-IgLON5 disease in the context of autoimmune or neurodegeneration theories?

Response 3: The main reference characterising movement disorders in IgLON5-IgG disease is ‘Gaig C, Compta Y, Heidbreder A, Marti MJ, Titulaer MJ, Crijnen Y, et al. Frequency and Characterization of Movement Disorders in Anti-IgLON5 Disease. Neurology. 2021;97(14):e1367-e81’. This study highlights that movement disorders in IgLON5-IgG disease are heterogenous, complex, and frequently present in combination with multiple movement disorders. Gaig et al. identify a subgroup of patients with a ‘PSP-like’ phenotype characterised by gait disorder combined with oculomotor abnormalities, all of whom had a chronic onset. We have stated that, “The subset of patients in keeping with a neurodegenerative picture generally present in a more non-inflammatory and chronic manner with a propensity for the PSP-like and cognitive impairment phenotypes” (page 5, lines 211-214). At this stage, it is unclear whether the other observed movement disorders like chorea, cerebellar ataxia, and facial/abdominal dyskinesias restrict with a patient subgroup based on primary driving mechanism. Ultimately, however, the model we propose is that patients with IgLON5-IgG disease may exhibit features of either a primarily neurodegenerative process, a primarily autoimmune process, or a combination of both. For example, a chronic onset process of neurodegeneration presenting with specific movement disorders may then be augmented by exposure to an inflammatory insult, such as a virus infection, triggering an autoimmune process and altering disease course. The heterogeneity of movement disorders in IgLON5-IgG disease is indeed complex and does appear to be, at least in part, explained by whether the primary driving mechanism is more pro-inflammatory and acute/subacute versus more non-inflammatory and chronic; however, further observational cohort studies are certainly needed to elucidate this connection and patients may exhibit features of both processes during their disease course.

We have revised the manuscript to improve the clarity surrounding this point: “Ultimately, the model we propose is that patients with IgLON5-IgG disease may exhibit features of either a primarily neurodegenerative process, a primarily autoimmune process, or a combination of both. For example, a chronic onset process of neurodegeneration presenting with a specific clinical phenotype may then be augmented by exposure to an inflammatory insult, such as a virus infection, triggering an autoimmune process and altering disease course. The heterogeneity of IgLON5-IgG disease is complex and further observational cohort studies are essential.” (page 4, lines 173-179).

Comments 4: Ref 8 and Ref 31 are redundant. 

Response 4: The in-text citations and reference list have been amended to remove the repeated reference. ‘Grüter T, Möllers FE, Tietz A, Dargvainiene J, Melzer N, Heidbreder A, et al. Clinical, serological and genetic predictors of response to immunotherapy in anti-IgLON5 disease. Brain. 2023;146(2):600-11.’ now appears once as reference 8.

Round 3

Reviewer 2 Report

Comments and Suggestions for Authors

1. An often discussed issue in neurodegeneration is selective vulnerability. (ref: Fu, H., Hardy, J. & Duff, K.E. Selective vulnerability in neurodegenerative diseases. Nat Neurosci 21, 1350–1358 (2018). https://doi.org/10.1038/s41593-018-0221-2). Perhaps the authors could also discuss the primary neurodegenerative mechanism of IgLON5-IgG in this perspective. Is there any relationship between vulnerable brain regions (such as hypothalamus and brainstem tegmentum) and the regional level of IgLON5 expression?

2. line 56, “3R and 4R isoforms”. The abbreviations “3R” and “4R” are first used here without more detailed explanation (but explained in the next paragraph).

3. The Ref.32 and Ref.37 appear to be incomplete. (Please also check other references) 

Comments on the Quality of English Language

Minor editing of English language required

Author Response

Thank you very much for taking the time to review this manuscript. We appreciate your feedback. Please find the detailed responses below and the corresponding revisions in track changes in the re-submitted files.

Comments 1: An often discussed issue in neurodegeneration is selective vulnerability. (ref: Fu, H., Hardy, J. & Duff, K.E. Selective vulnerability in neurodegenerative diseases. Nat Neurosci 21, 1350–1358 (2018). https://doi.org/10.1038/s41593-018-0221-2). Perhaps the authors could also discuss the primary neurodegenerative mechanism of IgLON5-IgG in this perspective. Is there any relationship between vulnerable brain regions (such as hypothalamus and brainstem tegmentum) and the regional level of IgLON5 expression?

Response 1: Thank you for this suggestion. We have revised the manuscript (page 2, lines 62-71) to include the following commentary:

“Fearnley, et al. examined the spatiotemporal expression of all IgLON family members in the developing murine nervous system at prenatal and postnatal stages (14). IgLON5 expression was detected prenatally in the eye and olfactory system with subsequent postnatal downregulaion at both sites, throughout development in the cerebral cortex, hippocampus, entorhinal region, habenula, and various nuclei of the thalamus, and postnatally in the cerebellum and spinal cord (14). This contrasts postmortem pathology-mapping, which is largely based on end-stage pathology such as neuronal loss; partial overlap with the regions of interest identified by murine modelling of IgLON5 expression is noted and may explain at least part of the regional vulnerability observed in IgLON5-IgG disease (1, 14, 15).”

Comments 2: line 56, “3R and 4R isoforms”. The abbreviations “3R” and “4R” are first used here without more detailed explanation (but explained in the next paragraph).

Response 2: We have revised the manuscript so the abbreviations “3R” and “4R” are defined when they are first introduced (page 2, lines 58-59).

Comments 3: The Ref.32 and Ref.37 appear to be incomplete. (Please also check other references) 

Response 3: All references have been generated using EndNote software in accordance with IJMS formatting and checked manually to ensure no missing information.

In addition to the above revisions, we have included a new reference to expand upon the potential pathogenic mechanisms exerted by IgLON5-IgG (page 3, lines 136-140) (Landa, J.; Serafim, A. B.; Gaig, C.; Saiz, A.; Koneczny, I.; Hoftberger, R.; Santamaria, J.; Dalmau, J.; Graus, F.; Sabater, L., Patients' IgLON5 autoantibodies interfere with IgLON5-protein interactions. Front Immunol 2023, 14, 1151574.). This amendment reads:

“In keeping with this, Landa, et al. identified another potential mechanism by which IgLON5-IgG isolated from patient sera may exert a pathogenic effect [38]. Firstly, they demonstrated that physiologic IgLON5 undergoes spontaneous ectodomain shedding and interacts with other members of the IgLON family and, secondly, that both IgLON5-IgG1 and IgG4 subclasses disrupt this interaction [38].”